# Communicating Branching Plans for Human-Agent Decision Making

## Julie Porteous[1], Alan Lindsay[2], Fred Charles[3]

[1] School of Computing Technologies, RMIT University, Melbourne, Australia
[2] School of Mathematical and Computer Sciences, Heriot-Watt University, Edinburgh, UK
[3] Faculty of Science and Technology, Bournemouth University, Poole, UK
julie.porteous@rmit.edu.au, alan.lindsay@hw.ac.uk, f.charles@bournemouth.ac.uk

## Abstract

Recent advances in visualisation technologies have opened up new possibilities for human-agent communication. For systems where agents use automated planning, visualisation of agent planned actions can play an important role in allowing human users to understand agent intent and to help decide when control can be delegated to the agent or when they need to be involved. We are interested in application areas where branched plans are required, due to the typical uncertainty experienced. Our focus is how best to communicate, using visualisation, the key information content of a branched plan. It is important that such visualisations communicate the complexity and variety of the possible executions captured in a branched plan, whilst also connecting to the practitioner's understanding of the problem. Thus we have developed an approach that: generates the complete branched plan, to be able to provide a full picture of its complexity; a mechanism to select a subset of diverse traces that characterise the possible executions; and an interface that uses 3D visualisation to communicate details of these characterising execution traces to practitioners. Using this interface, we conducted a study evaluating the impact of different modes of presentation on user understanding. Our results support our expectation that visualisation of characterising branched plan execution traces increases user understanding of agent intention and range of plan execution possibilities.

## Introduction

In systems requiring joint human and AI agent decision making there is a need for human users to understand the intentions of agents, along with the agent rationale for different decisions. This requires the AI agent to be able to explain its reasoning to the human, something which remains a significant challenge (Fox, Long, and Magazzeni 2017; Miller 2019). This is reflected in initiatives like DARPA's Explainable AI Program (Gunning and Aha 2019) and events such as (EXTRAAMAS 2020; XAI 2020; XAIP 2020).

For those application domains where AI agents (virtual or robot) use automated planning to control behaviour, the challenge is how to clearly communicate to the human the intentions of the agent which are encapsulated in its generated plans. It has been shown that 3D visualisation and simulation of agent plans can help human user understanding of agent intent (Chakraborti et al. 2018; Zolotas and Demiris 2019). However, generating understandable visualisations is

challenging because a plan sequence already implicitly encapsulates the balance made between dependency, constraint and choice, as well as the implied implementation of the plan steps themselves. This challenge is exacerbated when more complex plan structures are required, such as branched plans for partially observable domains due to inherent uncertainty.

In such contexts, the advantage of branched plans is that they allow efficient action sequences to be captured for each of the possible worlds that might occur and thus capture a diverse space of alternative solutions. However, the size of the space of possible executions makes it challenging to communicate this to a practitioner, along with the intentions of the agent whose behaviour is underpinned by the plan (e.g., Figure 1, the branched plan used in our evaluation). Thus, the problem we address in this work is how to communicate key information content of branched agent plans to human decision makers. This comprises: (i) how to communicate the complexity and variety of the possible executions captured within a branched plan; (ii) how to select subsets of execution traces that capture the scope of possibilities in branching plan structures to communicate to practitioners - something which is essential, as it is not desirable, or possible, to present all linearisations of a branched plan; and (iii) how to communicate the complexity of the branched plan and the selected execution traces, in ways that connect with their understanding of the problem.

To address these sub-problems, we have developed an approach that: (i) generates a full branched plan by branching on sensor action values and emphasizes key action points; (ii) selects subsets of execution traces that capture the scope of possibilities in branching plan structures; and (iii) demonstrated increased user understanding of agent intention and plan execution possibilities resulting from characterising diverse trace informed visualisation. The contribution of this work lies in the selection and adaptation of appropriate visualisations for human-agent branched plan communication.

## Background

A partially observable planning problem, e.g., (Bonet and Geffner 2011), can be defined by a tuple, $P = \langle F, A, M, I, G \rangle$, with fluents $F$, actions $A$, sensor model $M$, the initial state clauses $I$, over $F$, and goal, $G$. The clauses of the initial state provide both the known positive and negative literals, as well as constraints over the currently un-

known parts of the initial state. An action is defined by its preconditions and effects. An action is applicable if its preconditions are satisfied in the agent's partial state and the application of an action causes its effects to be applied to the agent's current state. Sensing actions are triggered whenever they become applicable and their observations update the agent's state. A solution to the problem is a branched plan, $\pi$, which has both deterministic action nodes and branching nodes, such that every branch of the tree results in a goal state. The branching nodes are labelled with a proposition and have branches for each of the possible valuations. The application of a branched plan requires traversing the plan tree applying the deterministic actions and selecting the appropriate branches by detecting the value of the proposition in the environment.

Partially observable planning problems, $P$, with a certain subset of exclusive-or knowledge can be compiled into deterministic classical planning problem, $P_{DET}$, following the approach in (Bonet and Geffner 2011). A key aspect of this encoding is that each sensing action is replaced by a pair of standard actions: one captures the effect of the sensor in the case that its proposition holds in the world and the other for the negative case. As a result the valuation of the sensors becomes a choice for the planner to make. A solution for $P_{DET}$ is therefore an optimistic and partial solution for $P$, which we denote, $\pi_{optimistic}$.

## Virtual Construction Domain

As a test bed for this work we have developed a virtual construction domain and used this setting for the development of the 3D visual interface. The planning domain captures various typical aspects of construction, including preparation, movement of robots and materials and the actual construction itself. Scenarios feature uncertainty in both the required preparation of the ground to permit construction and movement, and in the integrity of building materials. The planning domain model includes actions for movement, block-placing and sensing actions for identifying debris, such as rubble and rocks, in the environment. We have also developed a virtual environment for presenting 3D visualisations of branched construction plans to practitioners. We use this as a running example throughout the paper.

## Branched Plan Generation

We are interested in application domains that require branched plans, due to the uncertainty that is experienced. Branched plans allow efficient action sequences to be captured for each of the possible worlds that might be encountered during execution (with respect to the model). As a result branched plans can capture a diverse space of alternative solution sequences. Moreover sensor values and traces are not associated with likelihoods, leading to an interpretation that each of the executions is as likely as any other.

Given our focus in this work, we require a complete branching tree structure to be generated, so that we can provide a practitioner with a full picture of the complexity of the branched plan. Our approach to this generation builds on the K-Replanner (Bonet and Geffner 2011): an online

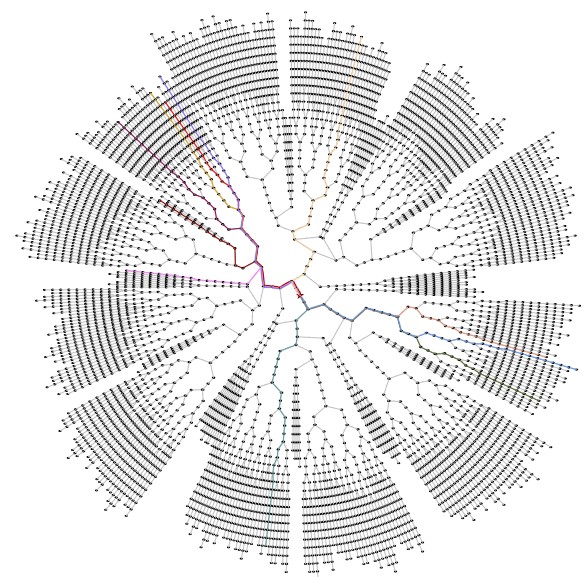

Figure 1: Visualisation of the branched plan for the problem used in the evaluation (see text for further detail).

approach to partially observable planning which supports efficient plan generation through a compilation to classical planning. The underlying classical planner is used to generate an optimistic plan, $\pi_{optimistic}$ and the K-Replanner approach follows this plan until an inconsistency is discovered, at which point it replans. However, as K-Replanner only explores individual real worlds, we extend it to generate the full plan, following the approach of (Komarnitsky and Shani 2016).

At each sensing action encountered in the optimistic plan, the plan is branched for each of the possible values and each of these branches is explored iteratively. Although not explored in this work, we observe that if the entire tree is prohibitively large it would be possible to explore a partial plan, by bounding the number of branching points. The action sequences from this partial plan could be bounded and used as input for our visualisation tools (this is particularly suited to one of our presentation modes, referred to as INTERLEAVED mode, as discussed in section "Empirical Evaluation").

Also, the K-Replanner's optimistic plan, $\pi_{optimistic}$, plays an important role in our approach to communicating, through visualisation, the branched plan possibilities to a practitioner. As this optimistic plan is used to drive the planner's strategy during planning it means that our visualisation is consistent with the intention of the system. Whereas with more direct approaches to partially observable planning it might prove challenging to accurately mirror their strategies, K-Replanner's strategy is particularly amenable. For further detail on optimistic plan visualisation see section: "User Interface: Presenting 3D Visualisations".

## Selecting Characterising Plan Traces

Our focus is communicating branched plans to human practitioners, however it is not typically possible, or desirable, to present all linearisations. Thus we propose using a small

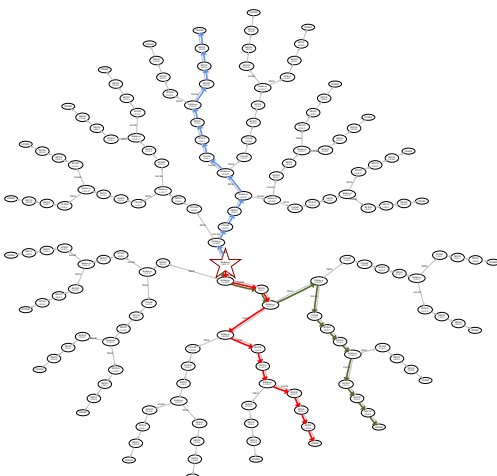

Figure 2: Radial visualisation of branched plan for sample problem: indicating actions and branching points; extended with selected execution traces (highlighted, view in colour).

subset of the execution traces to provide examples from across the broad scope of the alternatives captured in the plan. The intention is to provide the user with the intuition of what is captured within the plan without the burden of fully examining every trace. We are therefore interested in selecting a small number of execution traces that characterise the range of executions represented by the branched plan.

Our selection mechanism uses a dissimilarity measure to estimate the difference between two alternative action sequences. Using this measure on action sequences ignores other features in the trace, and means that the similarity of two traces is determined by the difference in the agent's actions and not differences in such things as sensor readings, which can be irrelevant. Using the dissimilarity measure allows the full set of linearisations to be clustered to identify its key groupings. Clustering provides flexibility, allowing a balance between the number of clusters and loss of detail.

**Dissimilarity Measure**  To estimate the dissimilarity between two execution sequences we use the Levenshtein distance (Levenshtein 1966): the distance between two word sequences which provides a measure of the edit difference between the sequences, while also respecting ordering. In our case we use unique words for each ground action. This measure was used as it is directly applicable to partially observable planning domains and it has been demonstrated that the approach leads to the identification of diverse plans (Coman and Munoz-Avila 2011).

**Clustering Execution Traces**  Clustering identifies groups of elements, which are similar (or close) to the elements in their own group, while being dissimilar (or far away from) elements in other groups. We therefore aim to break the space of possibilities into clusters, each representing similar execution traces.

We used the Partitioning Around Medoids (PAM) implementation of the $k$-medoids method (Kaufmann and Rousseeuw 1987). This approach partitions the data into $k$ clusters, each associated with a representative data point (the *medoid*), considered the most central in the cluster. This approach operates from a dissimilarity matrix, which can be computed by comparing each pair of traces using the dissimilarity measure. The medoids are central members of their respective clusters, and so we use them as the representatives of their clusters. For an appropriate value of $k$, this set of medoids will identify diverse execution traces, characterising the execution traces in the plan.

**Selecting the Number of Clusters**  The appropriate choice of $k$ is likely to depend on the application domain and the depth of understanding that is appropriate for the user, which might relate to the seriousness of inappropriate action, e.g., safety and security concerns. We therefore prefer the relatively lightweight $k$-medoids algorithm, which provides the necessary flexibility. One way to select a reasonable value for $k$ is to calculate the average *silhouette* score for the clusters (Rousseeuw 1987), which evaluates the clusters by averaging the similarity within clusters and dissimilarity between clusters, with respect to the distance measure. This can only be evaluated accurately for at least 2 clusters, so we first test to determine whether more than 1 cluster is appropriate (Duda, Hart et al. 1973). The optimal average silhouette score indicates a good trade-off between the size of $k$ and the amount of dissimilarity in each cluster. We therefore see this as a suitable default value.

# 1. Communicating Branched Plan Possibilities

The size of branched plans makes it challenging to communicate the space of possible executions to practitioners and consequently the intentions of agents whose behaviours are underpinned by such a plan. Our approach to address this exploits two visualisation methods to: (i) communicate the space of alternative execution traces captured in the plan; and (ii) highlight the set of characterising execution traces (selected using the approach discussed earlier).

## (i) Communicating Space of Alternative Executions

We use a visualisation of the complete space of alternative executions in order to provide an indication of the number and complexity of the possible alternatives. Although there is no intention that this will lead to an understanding of the actual traces, we extend the visualisation to allow practitioners to explore the possible executions in some detail.

Branched plans allow efficient solutions to be captured for each of the possible concrete states. For our branched plan generator, the branching nodes are associated with sensing actions allowing an appropriate course of action to be selected for each sensor valuation. In scenarios with a small amount of uncertainty the plan might be captured by a concise tree. As the level of (relevant) uncertainty increases, the size of the tree will grow and in some cases the tree will be very large. We observe that there is a natural similarity between a branched tree and a classical planning state space. This allows us to build on recent results in state space visualisation (Magnaguagno et al. 2017, 2020) to effectively communicate the size and complexity of the branched plan.

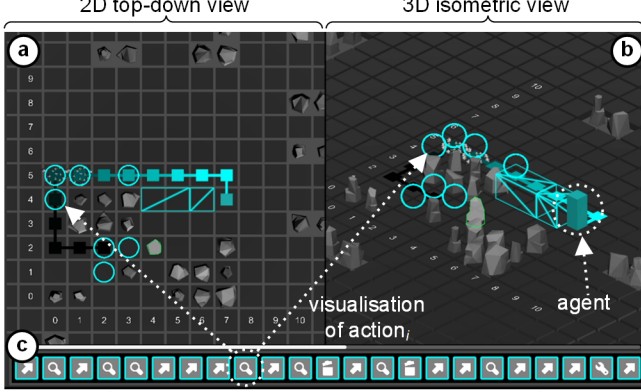

Figure 3: Graphical user interface: top-down (a) and isometric (b) simultaneous views of agent action 3D visualisation (simulation); sequence of agent actions (c) displayed as a timeline using icons (see text for detail).

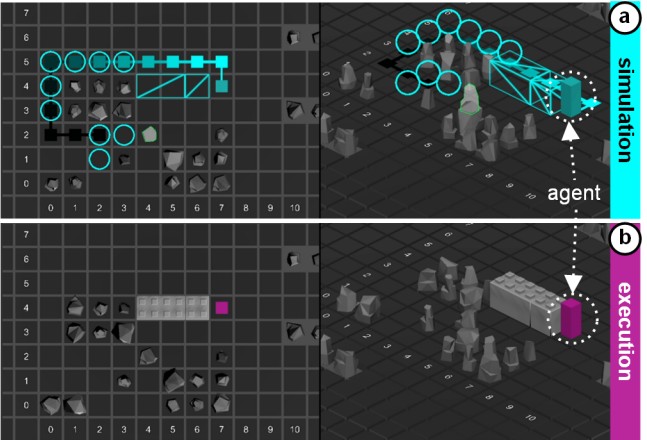

Figure 4: Visualisation example showing difference between: (a) *simulation*; (b) *execution* of selected plan traces.

We use a radial layout to visualise the tree. The root of the tree naturally sits in the centre of the visualisation and the branches of the tree expand from the root outwards. The graph has action and sensor action nodes and edges from sensor actions are labelled with the associated sensor value (i.e., $True$ or $False$). An example visualisation of the branched plan for a small construction problem is presented in Figure 2 and the branched plan for the problem used in our evaluation is presented in Figure 1. Whereas in (Magnaguagno et al. 2017, 2020) the search spaces branch on alternative choices, our plans branch on the valuation of sensor actions. It is therefore appropriate that the distance from the centre reflects the length of the execution, so that clearly longer executions can easily be identified.

We have extended visualisation by emphasising the key action nodes, i.e. those that achieve subgoals (shown with emboldened border in Figures 1 and 2). In order to reduce

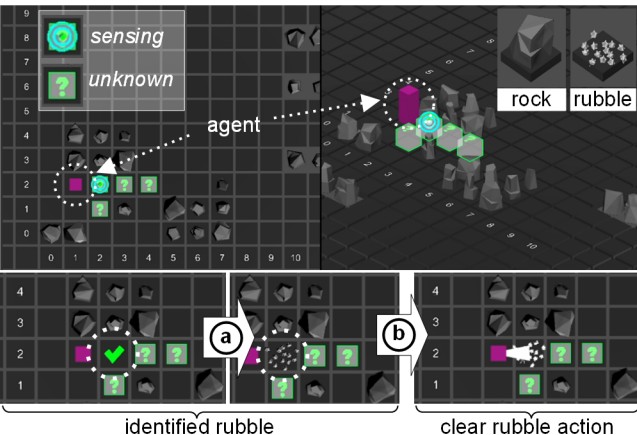

Figure 5: Example *sensing action* with agent in (1,2) sensing in (2,2). Here, the agent identifies rubble (a) and clears it (b).

the complexity of the visualisation, the nodes are annotated with simplified representations of the actions and sensor actions. Further information is provided through tooltips, which provide longer descriptions of the actions and decision points, as well as key state information.

### (ii) Highlighting Characterising Execution Traces

We have used these selected execution traces to enhance the branched plan radial visualisation and provide meaningful guidance to assist practitioners to navigate the tree and understand its alternatives. Figure 2 illustrates the visualisation of diverse alternative execution traces for our construction world, with differently coloured lines added to the radial plan visualisation for each diverse trace. Importantly, these characterising execution traces are the ones communicated to the practitioner, using 3D visualisation, as discussed next.

## 2. Communicating Selected Execution Traces

The 3D visualisation of a characterising execution trace provides an effective mechanism for clearly communicating the agent intention captured in that particular trace. We contrast this *3D visualisation* of an individual trace to the *radial visualisation* of a branched plan discussed earlier, as it refers to the use of 3D graphics to provide a visual run through of the sequence of actions, via animations, in a virtual environment. This 3D visualisation can be either: prior to execution, i.e. *simulation*; or the actual *execution* itself.

We have developed a graphical user interface for presenting such 3D visualisations to practitioners. The interface is implemented using the Unity3D game engine. An example is shown in Figure 3. It provides side-by-side synchronous views of the current action being visualised: a top-down view of the agent acting in the world (left-hand side), and a 3D isometric view (right-hand side). In addition, the interface has an icon-based representation of the sequence of actions in the execution trace (along the bottom). The icons are coloured as follows: grey if the action is yet to be visualised; yellow if currently being visualised; or green if fully visualised.

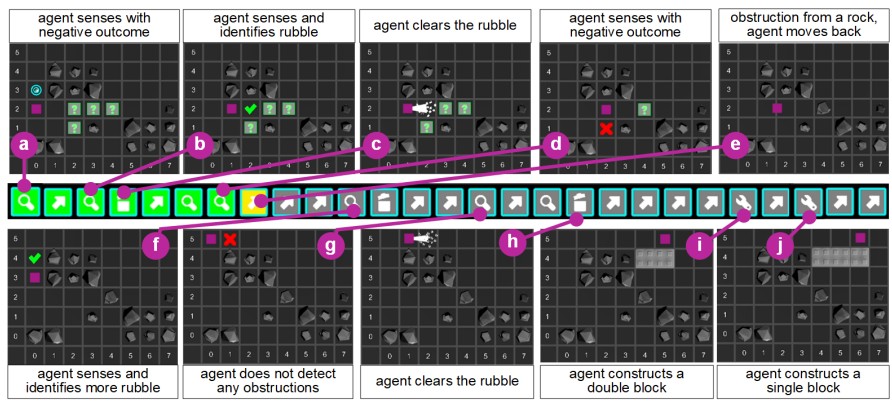

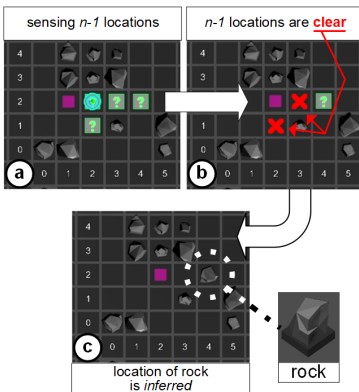

Figure 6: Example 3D visualisation of agent action sequence showing: a number of sensing actions (a, b, d, f, g); clearing of rubble (c, h); and construction (i, j).

Figure 7: Inference example (see text).

## 3D Visualisation of Action Sequences

The generation of 3D visualisations of agent actions relies upon the ability of the interface to convey realistic and semantically meaningful animations within the virtual environment. To maximise the practitioner's understanding of the execution traces that are visualised, we created contextually identifiable discrete sets of 3D animations constituting meaningful representations of agent actions.

An important requirement for the 3D visualisation is to provide visual representations that clearly differentiate between *simulation* and *execution*, and to ensure animations are consistent and graphically similar. Thus, actions use the same graphical animations in simulation and execution but use a different rendering style: simulation actions are rendered in turquoise-coloured wireframe render; whilst execution actions are rendered in fully-shaded grey (Figure 4).

A key concept in branched plan generation is the use of sensing actions (see section "Background"), which must be visualised so that practitioners can understand the implication of the valuation process. For instance, as shown in Figure 5, it might be that if rubble is identified the agent's plan is to clear the rubble before proceeding further.

As an illustration of 3D visualization of an agent sequence, as shown to a practitioner, Figure 6 shows a full trace of an *execution* of actions. The figure includes several sensing actions, actions clearing rubble and construction.

## 3D Visualisation of Inference Rules

Our approach to branched plan generation allows for agent inferences about locations of certain objects within the domain, as reported by (Bonet and Geffner 2011). For our application, inference rules are linked to agent sensing actions

with respect to "exclusive-or" knowledge about locations suspected to contain rubble or rock. For 3D visualisation, such locations are shown with a question mark (grey square, green question mark), as in Figure 7(a). Over a series of sensing actions the agent gathers more information about locations and the results are visualised, with question mark replaced by either a representation of the object sensed if *True*; or a red cross for *False* (e.g. sensing actions in Figure 7).

## User Interface: Presenting 3D Visualisations

An important aspect of eXplainable AI Planning (XAIP) is helping users understand what decisions have been made in a plan, and why. As it is not always feasible to present all branched plan linearisations, our approach is to select a diverse set of execution traces, that characterise the possibilities captured within the full branched plan, and to present these to the practitioner via a series of 3D visualisations within the user interface. Here, we describe the ways in which presentations can be organised to exploit plan structure, as appropriate, to assist practitioners understanding by giving some transparency to the agents' planning strategy.

## Consistency with the Agent's Intent

Building on (Bonet and Geffner 2011), our approach to branching plan generation is based on a specific strategy: construct the branched plan starting from a single optimistic plan, $\pi_{optimistic}$, and iteratively branch for alternative sensor values. We mirror this planning strategy in our approach to presenting execution visualisations: we use this optimistic plan as the "backbone" for presentation of 3D visualisations.

Thus, within the user interface, both the agent's optimistic plan and the selected set of characterising plan traces can be presented to the practitioner – in order to establish common ground between the practitioner and the agent (the agent could follow one of the diverse plans; however, because it is not used to guide plan construction it may lack the rationality and focus of the optimistic plan).

## User Interface Modes

The appropriate mode of presentation of information will often vary between applications. For example, in some scenarios, where the practitioner must have complete understanding of the agent plan before execution starts (e.g., high risk applications), *simulation* can be used to safely explore different possible plan alternatives prior to execution. Whereas in other situations it might be appropriate to interleave the presentation of information, via *simulation*, throughout the *execution* (e.g., relatively slow execution applications). We observe that during actual agent execution the uncertainty in the concrete state will reduce, isolating a smaller portion

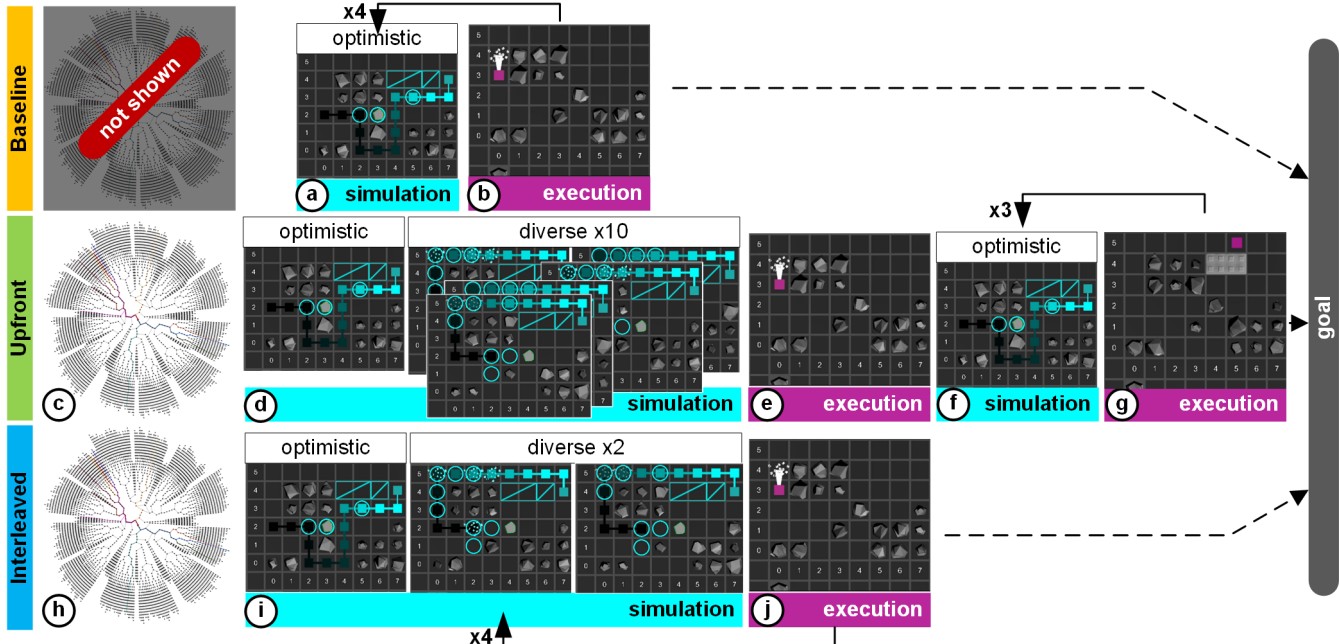

Figure 8: User Study Experimental Conditions. Participants are shown (planning instance with 4 breakpoints): BASELINE to each breakpoint/goal, simulation of optimistic plan only (a), and execution (b); UPFRONT radial visualisation (c), then, to first breakpoint, simulation of k diverse traces and optimistic plan (d), execution to the first breakpoint (e), then, to each remaining breakpoint/goal, simulation of optimistic plan only (f), and execution (g); INTERLEAVED radial visualisation (h), then to each breakpoint/goal, simulation of k diverse traces and optimistic plan (i), and execution (j). See text for further details.

of the overall branched plan. Therefore, where appropriate, presenting information at stages during execution can provide sets of execution continuations that are more focused towards the unfolding execution. Thus, the user interface was developed with the following modes of presentation:

- *Simulation:* presentation of 3D visualisations of selected traces from the full branched plan prior to execution. In simulation mode, each trace implies a set of assumed sensor valuations for the sensing actions, which are used to generate the appropriate visualisations (differentiated using a different render mode as discussed earlier).
- *Execution:* presentation of 3D visualisation of the actual execution from a concrete starting state through to either a *breakpoint*, or the final goal. A breakpoint is a point where the actual sensor valuations differ from the assumed sensor valuations in the plan trace being executed.

We note that breakpoints provide a useful opportunity to supply further information to the practitioner, and is something we explored in our user study (next section).

## Empirical Evaluation

For evaluation we developed a prototype interface for our construction world (see section "Background") featuring:

- Branched plan generation, based on the K-Replanner, extended to output the full contingency tree.
- Selection of a set of execution traces characterising the scope of possibilities within generated branched plans.

- Radial visualisation of branched plans, using an approach building on (Magnaguagno et al. 2017, 2020)
- Virtual environment for presentation to practitioners via 3D visualisation of selected traces.

## User Study

We set up a user study to investigate the impact of different modes of information presentation and exploration on understanding of agent plans, in terms of awareness of agent intended actions under uncertainty, the agents' overall goal and participant ratings of preparedness to answer questions.

We recruited 24 native english speakers to participate in the study. The participants were not experts in planning. The study was delivered via an online questionnaire and all participants were provided with an introduction to the setting and aspects of the visualisation e.g. the visual difference between *simulation* and *execution*, as in Figure 4.

For the study a single planning instance was used, with 4 breakpoints, i.e., points where, at execution, the actual sensor values differ from those assumed in the optimistic plan.

**Experimental Conditions** Participants were randomly assigned to one of three conditions which differed with respect to: (i) whether the range of branched plan possibilities were communicated i.e. shown radial visualisation of the full branched plan; (ii) the number of selected branching plan traces they were shown in *simulation mode*; and (iii) the timing of visualisation of branched plan simulations in

| Q1: | *"... sufficient information about the possible alternative executions for you to anticipate the execution steps ..?"* |
|---|---|
| Q2: | *"... what do you think the agent will do next?"* |
| Q3: | *"How confident are you about your answer?"* |
| Q4: | *" ... do you know what the agent's goal was? ... How confident are you? ... State what the agent's goal was"* |
| Q5: | *"... How well prepared were you ..?"* |

Figure 9: Questions for user study: Q1-Q3 were asked after each breakpoint; Q4-Q5 were asked at the end of the study.

| P | Q1: Information | | | Q2: Next Action | | | Q3: Confidence | | |
|---|---|---|---|---|---|---|---|---|---|
| | U | I | B | U | I | B | U | I | B |
| 1 | 4.6 | 2.5 | 2.1 | 62.5 | 62.5 | 62.5 | 3.8 | 2.8 | 2.6 |
| 2 | 4.5 | 3.1 | 2.1 | 75 | 62.5 | 75 | 3.8 | 2.8 | 2.2 |
| 3 | 4.3 | 3.1 | 3.2 | 25 | 87.5 | 75 | 3.8 | 2.3 | 3.1 |
| 4 | 3.7 | 2.5 | 3.7 | 62.5 | 87.5 | 100 | 3.7 | 2.1 | 3.5 |
| $\mu$ | 3.6 | 2.8 | 2.8 | 56 | 75 | 78 | 3.8 | 2.5 | 2.8 |

Figure 10: Breakpoint Question Responses: Breakpoint (P), UPFRONT (U), INTERLEAVED (I), BASELINE (B). **Q1** (6-point Likert scale: 0=Not at all, 5=Yes Fully): results to be expected: UPFRONT and INTERLEAVED receive more information earlier and yield higher rankings; for BASELINE, rankings increase over execution. **Q2** (% correct): Results show similar avg. performance for INTERLEAVED and BASELINE with poorer performance for UPFRONT. **Q3** (6-point Likert scale: 0=No idea, 5=High): Results show high confidence for UPFRONT with similar avg. confidence for INTERLEAVED and BASELINE. See text for details.

relation to actual *execution*. These conditions are as follows and illustrated in Figure 8:

- BASELINE: participants were shown *simulation* of the optimistic plan (a), followed by *execution* of the optimistic plan to the next breakpoint (b)), or the goal. After each breakpoint this continued, with *simulation* of the optimistic plan from the current state, followed with *execution* of the optimistic plan looping through to the goal.

- UPFRONT: participants were shown the radial visualisation of the full branched graph (c). Then *simulation* of each of $k$ diverse traces (rationale for $k$ below) and the optimistic plan (d), followed by *execution* of the optimistic plan through to the first breakpoint (e). Continuation from the first breakpoint, repeatedly loops, showing *simulation* of the optimistic plan from the current state (f), followed by *execution* of the optimistic plan to the next breakpoint or goal (g), repeating through to the goal.

- INTERLEAVED: participants were shown the radial visualisation of the full branched graph (h). Then *simulation* of each of $k$ diverse traces (rationale for $k$ below) and the optimistic plan (i), followed by *execution* of the optimistic plan, through to the next breakpoint or the goal (j)). Continuation from each breakpoint repeatedly loops starting from the optimistic plan from the new current state.

**Rationale for values of k** For INTERLEAVED $k$ was 2, the mode of the silhouette scores across the breakpoints. It

| | Q4: Goal Awareness | | | Q5: Preparedness |
|---|---|---|---|---|
| | G | C | A | |
| UPFRONT | 100% | 4.1 | 87.5% | 3.88 |
| INTERLEAVED | 75% | 2.8 | 87.5% | 3.25 |
| BASELINE | 37.5% | 2.8 | 62.5% | 2.88 |

Figure 11: Post-Execution Questions. Q4 Goal Awareness: % "yes" to know agents goal (G); Confidence (C); % correct goal awareness (A). Q5 Preparedness: (5 point Likert scale: 0=Not Prepared, 5=Well Prepared). Overall, results show increases in confidence, accuracy and preparedness rating for more informed users (UPFRONT and INTERLEAVED).

was decided not to use the silhouette scores directly, as this would mean presenting different number of plans at each breakpoint and would introduce too much variation into the study. For UPFRONT, $k$ was 10 to ensure these participant saw a similar number of videos overall to INTERLEAVED.

**Expectation** Our expectation was that the UPFRONT participants would have a strong sense of how the initial stages of *execution* would progress, but perhaps lose confidence towards the end, as the *execution* (perhaps) deviated from the plans they observed at the start. We expected that the INTERLEAVED participants would perform fairly well, but possibly feel less prepared in early stages, or where the *execution* deviated from the small collection of plans they observed.

## Results

**Breakpoint Questions** During execution, at each breakpoint, all participants were asked about the information they had seen so far, what they thought the agent would do next and their level of confidence. Text for these questions, Q1-Q3, is in Figure 9, and responses summarised in Figure 10.

For Q1, sufficient information, the more informed conditions, INTERLEAVED and UPFRONT, gave higher rankings than BASELINE, with respect to receiving sufficient information to anticipate execution steps they observed. This is to be expected, as it reflects these participants receiving more information prior to execution. Of interest is the increased rankings over breakpoints 1-4 for BASELINE. We suspect this results from clarification through exposure to execution visualisations in the absence of diverse trace simulations.

For Q2, participant understanding of agent intentions (Next Action), results show similar performance for INTERLEAVED and BASELINE (overall avg. 75% and 78% respectively) with poorer overall performance for UPFRONT (56%). However analysis of responses for breakpoint 1 are interesting: for this question all UPFRONT participants got the direction correct, in stark contrast to BASELINE where all incorrect responses indicated moving in the wrong direction. We observed similar behaviour at each breakpoint (in particular, breakpoint 3). This strongly suggests that the UPFRONT participants understood the intended direction, but some failed to understand the wireframe communication of clearing rubble (e.g. some participants noted, in the open text questions at the end of the questionnaire, that the visualisations had been unclear about whether rubble should be

removed or passed through). From this interpretation, the results are consistent with our expectation: there will be more variation on participant understanding of how the agent intends to proceed (in terms of direction) with BASELINE.

Q3, Confidence ratings, are high for UPFRONT (avg. 3.8) with similar overall confidence for INTERLEAVED and BASELINE. For UPFRONT, this is to be expected given the upfront simulation of diverse traces. Interestingly, despite this confidence for participants in UPFRONT, they weren't necessarily correct in their answers as shown for Q2 (avg. 56%), in contrast to higher correctness scores for INTERLEAVED and BASELINE.

**Post-Execution Questions**  Following completion of execution all participants were asked about their awareness of the agents overall goal and their overall feelings of preparedness (Q4-Q5 shown in Figure 9). Responses to these questions are shown in Figure 11. For UPFRONT this shows consistently high rankings of confidence and preparedness, in line with responses to execution questions. In contrast to the execution questions, users in this condition exhibited similar accuracy, with respect to the agent goal, to INTERLEAVED. This improvement suggests increase in level of informedness from exposure to the execution visualisation itself. The low rankings for Q4-Q5 for BASELINE are to be expected given they are less informed than the other conditions.

**Overall**  Our expectation was that UPFRONT would have a strong sense of how the initial *execution* would progress, but perhaps lose confidence as the *execution* (perhaps) deviated from the initial plans they saw. We expected that INTERLEAVED would perform fairly well, but possibly feel less prepared in early stages, or where *execution* deviated from the small collection of plans they had seen so far.

For UPFRONT participants performed beyond expectations, appearing to use what they had understood from visualisation at the outset to guide their predictions. However, for INTERLEAVED it appears that the number of plans was insufficient for them to interpolate accurately and confidently. This is an important result, which indicates that the silhouette score might not be appropriate, at least for beginners.

## Related Work

Explainable AI Planning (XAIP) (Fox, Long, and Magazzeni 2017), is an area of growing importance and focus in planning, which is motivated by the need for trust, interaction and transparency between users and AI controlled agents (Hoffmann and Magazzeni 2019; Kambhampati 2019; Lindsay 2019; Sohrabi, Baier, and McIlraith 2011). Communicating the intentions of the agent plays an important role in XAIP. The form of visualisation for communicating intention vary from annotations indicating objects that are involved in the agent's plan (Chakraborti et al. 2018), the indication of intended movements of the robot (Chadalavada et al. 2015) and a visualisation of the agent's internal decision making (Chakraborti et al. 2017a). A common approach is to use specialised visualisations to present the intentions of an agent to a system user, e.g., through projection (Leutert, Herrmann, and Schilling 2013; Chadalavada et al. 2015) or augmented reality (Chakraborti et al. 2018; Zolotas and Demiris 2019). In (Chadalavada et al. 2015) it is demonstrated that projecting the robots intentions improves the user rating of the robot. In (Chakraborti et al. 2018) they build a domain specific visualisation, using augmented reality to project a robot's intentions, which also allows manipulation. The approach for visualising the complete branched plan that we use here, is related to other domain independent visualisations (Magnaguagno, Pereira, and Meneguzzi 2016; Magnaguagno et al. 2017, 2020). In (Magnaguagno, Pereira, and Meneguzzi 2016) they present a plan visualisation, which exploits a visual metaphor in order to communicate abstract planning concepts, such as action preconditions. Our branched plan visualisation was inspired by the state space visualisation of (Magnaguagno et al. 2017).

(Chakraborti et al. 2017b) assume the user model is available so they can isolate specific situations where the user's understanding of the current sequence might fail. In this way they consider explanations as model corrections. We do not assume a user model, rather, that a common ground can be established by exploiting sequential plan visualisations.

Selecting or generating sets of diverse plans has been investigated in classical planning (Katz and Sohrabi 2020), and various applications have been identified, including risk assessment (Sohrabi et al. 2018) and user preferences (Nguyen et al. 2012). (Coman and Munoz-Avila 2011) adopt two diversity measures, including the measure we use, and demonstrate that both approaches lead to identifying diverse plans. They also demonstrate that a domain specific diversity measure was particularly effective. Our approach is compatible with any approach that can return a set of diverse plans (especially those able to vary set size).

## Discussion and Conclusion

In this work we have considered the problem of visualising a contingent plan and providing the user with visualisation of the intended plan and key information about the contingency tree. The aim is to provide access to the potential alternatives captured in the contingency tree, so users can better understand and assess time required and risk implied by the plan.

We have presented a general approach that provides a template for constructing branched plan visualisations. It focuses on general aspects of branched plans, such as diversity and inference, thus making it. generalisable to other settings e.g. using tools such as PLANIMATION (Chen et al. 2020).

The results of a user study assessing our approach are promising. They indicate that users can gain an awareness of agent intentions and the scope of alternative possibilities through exposure to selected traces which characterise the branched plan space.

In future work, the different modes of presentation and use of the silhouette score will be further explored, along with user preferences for different modes of presentation and the interface itself, especially in the context of in-situ visualisation of actions through mixed-reality, which will further support interactive exploration of the agents' plan traces.

## Acknowledgments

This work was partially funded by the ORCA Hub
(orcahub.org), under EPSRC grant EP/R026173/1; and
DSI Collaborative Grant CR-0016.

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
