# OpenReview forum: "Communicating Branching Plans for Human-Agent Decision Making"
_icaps-conference.org/ICAPS/2021/Workshop/XAIP — XAIP 2021_

### Official Review · AnonReviewer2 · 2021-06-29
**Nice paper, need to reify the contributions**

**Rating:** 7
**Confidence:** 4

**Review:**

This paper presents a system that can generate and visualize a branched plan to help users understand it. They generate a branched plan by considering sensing actions that may have different outcomes, and iteratively exploring all possible values. To communicate with the practitioners, they select a subset of the plan traces that sufficiently capture the diversity in plans by clustering plan traces using a dissimilarity score. The entire branched plan is shown as a radial tree, with the characterizing plan traces highlighted. Further, they create a 3D visualization system for a virtual construction domain and perform a user study to showcase the effectiveness of showing alternate plans from the branched tree versus just showing the optimistic plan, via the visualization.

The paper is well written and easy to follow. The included figures illustrate the capabilities of the visualization system well. The contribution of the paper is towards helping users understand branched plans, and while it is not explaining the planner decisions, I feel it falls under the purview of the XAIP workshop because it presents a user interface that helps bridge the communication gap between planning AI and human users.

The primary concern with this paper is the clarification of the contribution. From my current understanding, the contribution is towards the generation of branched plans and selection of characteristic traces, the presentation of which can be generalized to different visualization systems, but the paper claims the contributions to be related to visualization.

*"The contribution of this work lies in the selection and adaptation of appropriate visualizations for human-agent branched plan communication."*

If the primary contribution is the branched tree visualization, it has not been sufficiently explored in the user study, and there is no scenario that lets readers understand the contribution the branched tree visualization makes towards user understanding of the branched plan. It would be useful to include one such scenario, but that will probably require an additional study.
The user study is more focused on the 3D visualization of the construction domain, and how visualizing a different number of simulated plans at different branch points helps users understand the agent's intent. If the 3D visualization is the contribution, then this needs to be more clearly stated, as it is created for a specific domain, and there need to be some comments on its generalizability.

The authors note that it is challenging to generate understandable visualizations, which is a good point. It would be useful to see some of the design choices being evaluated against alternatives, or some discussion on how those choices were made to support the goal of finding the best way to visually communicate branched plans, as stated in the abstract.

**Minor comments:**

It is possible that some branches lead to unsolvable states. Is there a provision for visualizing or describing unsolvable branches in the designed system?

The plans are presented in accordance with the agent's intent, which is a good step towards transparency in Human-AI interactions. It would be useful to also include this information in the branched tree visualization somehow, like by labeling the optimistic plan

It would be helpful to have some more discussion on how the silhouette score was used to select the number of plans to show for the interleaved scenario. Why was the mode selected instead of the mean or median, as suggested earlier in the paper?

Were there any considerations made for people with different color perceptions or people viewing the visualization in black and white?

The tables in the results section have been labeled as figures. It would be better to label them as 'Table'.

For the user study, a description of the planning instance has been omitted. Were the participants told what the goal of the agent is? If it is possible, include a brief description in the text.

The term "contingency tree" is first used in the Empirical Evaluation section, without any introduction. This is minor, but it may be helpful to mention and define it earlier.

**Suggestions and Questions to the Authors:**
1. Address the issue of the specific contribution this work makes, whether it is algorithmic, visualization-based, or some combination. Highlight this in the text.
2. Give reasons for design choices if alternatives were considered.
3. Try to address and update the paper with the smaller changes suggested above in the "minor comments"

---

### Meta-Review · Area_Chairs · 2021-07-07

**Recommendation:** Accept
**Confidence:** 5

**Metareview:**

The paper looks at the problem of communicating and visualizing branching plans. This an important and unfortunately less-studied problem in XAIP and seems well suited for the workshop. The reviewer agrees and recommends accepting the paper, but also notes the lack of clarity about the primary contributions of the paper and the fact that the user study doesn't seem to test the stated primary contribution. I recommend accepting the paper. We recommend the authors clarify this in the paper and invite them to respond to the reviewer on the open review forum.

Also, we are working on getting one more review for the paper that will be posted after the notification date.

---

### Decision · Program_Chairs · 2021-07-08

Accept